# Rotational orientation control of a ground state ortho-H$_2$ dissociation on a metal surface

H. Chadwick ⓘ, G. Zhang ⓘ, C. J. Baker ⓘ, P. L. Smith & G. Alexandrowicz ⓘ ✉

When hydrogen molecules collide with a surface, they can either scatter away from the surface or undergo dissociative chemisorption. The relative probabilities of these different outcomes could depend on the rotational orientation of the impinging molecules, however, due to the lack of steric control techniques for ground state hydrogen, they could not be measured directly. Here, we demonstrate that magnetic field manipulation can be used to control the rotational orientation of H$_2$ molecules colliding with a nickel surface and change the balance between reactive and scattering collision events. Our measurements show that molecules which approach the surface while rotating within a plane parallel to the surface are less likely to undergo specular scattering than those rotating within a perpendicular plane. An opposite trend was measured for the likelihood of dissociative chemisorption. A possible link between these two findings, and its potential impact on the interpretation of dissociation mechanisms is discussed.

Controlling the quantum state of reagents in a chemical reaction, and in particular studying how the orientation of the molecule and its rotational plane (stereodynamics) affect the reaction rate, provides an ultimate test for our understanding of chemistry at the single molecule level[1]. Over the years this level of control has been achieved for several molecules using photo-excitation, trajectory deflection in electric and magnetic field gradients and velocity-orientation correlations in molecular beams (e.g., refs. 2–9). One molecule which could not be addressed by these controlled methods is H$_2$ in its vibrational and rotational ground state.

Beyond being the most abundant molecule in the universe, H$_2$ is the smallest and simplest molecular reagent, making it a challenging yet realistic system for accurate theoretical modelling of both gas phase and molecule-surface collisions, while also being important in numerous processes ranging from astrochemistry to the development of clean renewable energy[10–13]. 

Whilst the adsorption of H$_2$ on Ni(111) has been studied since the early days of surface science[14], our understanding of this system is far from complete, and there are discrepancies between state-of-the-art calculations and experimental data, as well as between various experimental data sets[15–17]. A common feature seen in different

experiments was that the measured sticking coefficients increase with the energy of the impinging molecules. However, the calculated values of the energy barrier to dissociation vary hugely between different DFT functionals, ranging from non-activated dissociation to high energy barriers, and none of these functionals is capable of reproducing the measured sticking coefficients across the full range of energies and incident geometries[15].

While the role rotational orientation plays in the collision of H$_2$ with a Ni(111) surface could not be studied experimentally, there are reasons to believe it could be important. For example, theoretical studies have predicted stereodynamic trends of H$_2$ dissociation on both flat and stepped copper surfaces[18,19], and pioneering experiments showed alignment of products in the associative desorption of D$_2$ molecules from Cu(111) and Pd(100) surfaces[20,21], which through detailed balance arguments suggests such trends should also exist in dissociative chemisorption. Furthermore, theoretical work has predicted a direct relation between the stereodynamics of scattering and dissociation for H$_2$ colliding with a cobalt surface[22]. However, it has previously not been possible to probe such correlations for H$_2$ experimentally. A counter argument for expecting significant stereodynamic effects, would be the existence of significant steering

Department of Chemistry, Faculty of Science and Engineering, Swansea University, Swansea, UK. ✉e-mail: g.n.alexandrowicz@swansea.ac.uk

mechanisms which guide the molecule towards an optimal reaction geometry. Steering has been predicted to be particularly important for collisions of low energy $H_2$ molecules with other surfaces[23–25]. If such steering is strong enough, it is often assumed that it will eliminate the dependence of the reaction rate on the rotational orientation of the impinging molecules[6], an assumption we will discuss later in the context of our experimental observations.

In this work we used a combination of inhomogeneous and homogeneous magnetic fields to manipulate the rotational projection quantum states of ground state ortho-$H_2$, i.e., the quantised rotational orientation of the molecules before they collide with a Ni(111) surface, and achieved control over the branching ratio of dissociative chemisorption of $H_2$ and scattering back into the gas phase. Our results reveal anticorrelated stereodynamic trends, where molecules approaching the surface in a helicopter-like geometry are more likely to react and dissociate whereas those approaching as cartwheels are more likely to undergo elastic scattering into the specular channel.

## Results and discussion

We studied the stereodynamic trends of $H_2$ molecules colliding with a Ni(111) surface using a magnetic molecular interferometer (MMI)[26,27] working in flux detection (FD) mode, which means we use magnetic fields to modulate the spin and rotational projection quantum state of the molecule before it hits the surface and follow how this changes the total intensity of a specific scattering channel. The MMI setup can be used to perform both FD measurements, as well as full interferometer (FI) measurements where both the incoming and outgoing quantum states are resolved[26–29]. While the wealth of information from a FI measurement is particularly beneficial to benchmark and test theoretical calculations[27,28], FD measurements can be much simpler to interpret, as will become obvious from the example we show below. An additional capability we have added to the setup is the use of mixed beams, where a small quantity of helium atoms are added to the molecular beam, allowing us to monitor changes in the surface coverage while simultaneously controlling the rotational states of the impinging hydrogen molecules, experiments which will be explained and presented later in this manuscript.

In the past FD measurements have been performed by using a strong depolarising field in the second arm of the apparatus[26] which results in a signal which is approximately independent of the final quantum state. For this work we modified the MMI setup, adding two differentially-pumped mass spectrometers at two different angles which allow us to measure the stagnated pressure of the scattered

beam directly, i.e., without passing the scattered beam through any state selecting components. Figure 1a shows a schematic of this configuration, whereas further details of the elements used are described in the methods section.

The basic principle of our control scheme exploits the fact a ground state ortho-hydrogen molecule has 9 different eigenenergies which differ in their nuclear spin and rotational projections $(m_I, m_J)$[30]. While the magnetic moments associated with the $m_I, m_J$ states of $H_2$ are 3 orders of magnitude weaker than those in paramagnetic, radical and metastable species[5,31,32], the subtle energy differences can still be used for state selective manipulation. More specifically, a hexapole polariser magnet[33] is used as a magnetic lens to create an initial population difference of quantum states in the beam. The hexapole magnet primarily selects the states based on their $m_I$ projections due to the large difference between the rotational and nuclear spin magnetic moments. The molecules then enter a perpendicular homogenous magnetic field, $B_I$, created by passing a current, $I_I$, through a solenoid coil, before entering the scattering chamber and colliding with the surface. The non-adiabatic field transition into the solenoid, results in the creation of a superposition quantum state, which generally has non-zero projections onto all 9 base functions. The superposition state evolves coherently in a reproducible way which depends on $I_I$, this allows us to control the projection of the quantum state onto any of the $m_I, m_J$ basis states[26]. The mass spectrometer illustrated in Fig. 1a measures the flux of $H_2$ molecules which undergo scattering into the specular channel for different total scattering angles ($\theta_{total}$) which can be set to either 45° or 22.5°.

The magenta markers in Fig. 2a show FD results for specular scattering of $H_2$ from a Ni(111) surface at a temperature of 500 K. The nozzle temperature was stabilised at 106 K producing a beam with a mean velocity of 1513 ms$^{-1}$ and a FWHM of 7%. Scanning the magnetic field clearly modulates the scattered signal intensity, i.e., the quantum state of the molecule before it hits the surface changes the specular scattering probability. As the nuclear spin orientation is not expected to change the scattering probability, the oscillations we observe should reflect the scattering stereodynamics, i.e., the fact that different rotational projection states have different probabilities of scattering into the specular scattering channel. The oscillation interference patterns measured in previous MMI studies, which included state selection after scattering, are sensitive to changes of both the magnitude and the phase of the molecular wave function during scattering, and their interpretation required fitting the elements of the scattering matrix[27–29]. As we shall show below, the FD

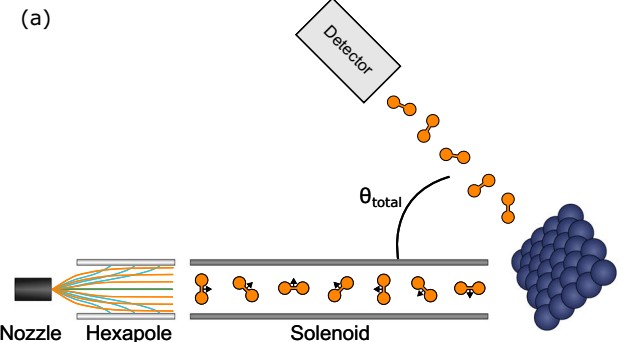

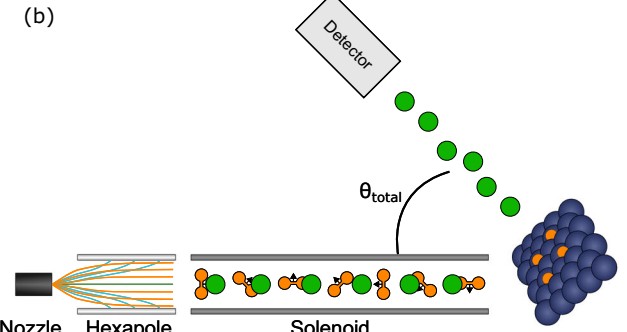

**Fig. 1 | Illustration of the measurement method. a** Schematic of the setup used for the FD measurements. The rotational orientation of the molecules which reach the surface is controlled by a combination of an initial magnetic lens (hexapole) followed by a perpendicular field $B_I$, generated by passing the control current $I_I$ through a solenoid coil. The controlled beam then collides with a surface. The flux of $H_2$ molecules scattered in the specular direction is measured by mass spectrometers,

positioned at $\theta_{total} = 45°$ or 22.5°. **b** To measure sticking stereodynamics a small fraction (10%) of helium (illustrated as green circles) is mixed in the $H_2$ beam. By switching between two currents in the solenoid ($I_I$), the relative populations of helicopter and cartwheel $H_2$ molecules reaching the surface are modulated. The flux of helium atoms scattered from the surface into a mass spectrometer is used to monitor the surface coverage and how it changes for different $I_I$ values.

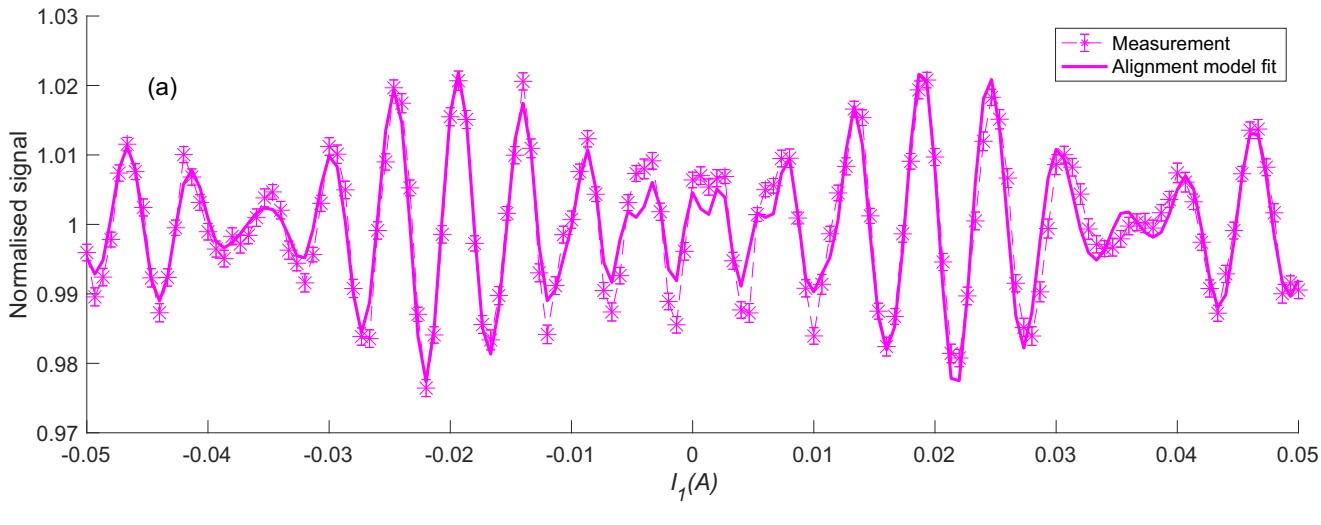

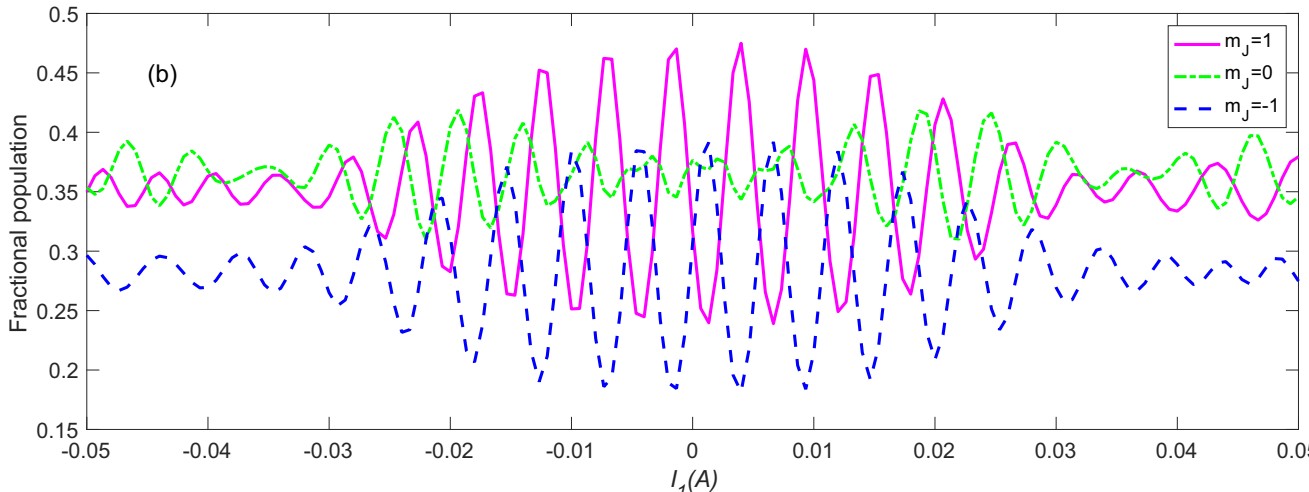

**Fig. 2 | Stereodynamic control of the specular scattering flux. a** The magenta asterisk markers connected by magenta dashed lines show the normalised signal of a flux detection measurement from a 500 K Ni(111) surface as a function of the solenoid current, $I_1$. The nozzle temperature was stabilised at 106 K producing a molecular beam with a mean velocity of 1513 ms⁻¹ and a FWHM of 7%. The error bars show the standard deviation of the values calculated from repeat measurements ($n = 61$). The thick magenta line shows the alignment scattering model for $\alpha = 1.5$, which fits the data almost perfectly. **b** The magenta (full), blue (dashed) and green (dashed-dotted) lines show the calculated average populations in the $m_J = 1$, −1 and 0 states of the beam arriving at the sample as a function of the control current $I_1$. The surface normal was used as the quantisation axis. Source data are provided as a Source Data file.

signal shown in Fig. 2a, which is insensitive to the final state of the molecule, allows us to employ a much simpler interpretation of the experimental results.

As the magnetic Hamiltonian for $J = 1$, $I = 1$ H$_2$ molecules is well known[30], we can use semi-classical calculations (see methods for more details) to propagate initially pure $m_I$, $m_J$ quantum states emerging from the end of the hexapole polariser and calculate the quantum state of the molecule just before colliding with the surface[26]. Figure 2b shows the average rotational projection populations of the beam particles as they arrive at the surface, using the surface normal as the quantisation axis, and summing over the three different $m_I$ states. The populations in the $m_J = 1$, −1 and 0 states as a function of the current $I_1$ are shown using solid magenta, dashed blue and dash-dotted green lines, respectively. In this paper, we use the common terminology of calling molecules in the $m_J = 1$, −1 states as the two (counter-rotating) helicopters, and those in the $m_J = 0$ state as cartwheels.

One observation we can make by looking at the populations plotted in Fig. 2b, is that the central ($I_1 = 0$) region is dominated by

strong oscillations between the populations of the two counter-rotating helicopter molecules, which means that if these two states scattered with significantly different probabilities we would expect a pattern which oscillates most strongly at the central region, which is not what we see in the experimental data. This follows our expectations, as due to the symmetry of the surface with respect to the scattering plane, we anticipate the sense of helicopter rotation to not affect the scattering. Next we note that the experimental data in Fig. 2a has a similar pattern to that of the calculated cartwheel population (green dash-dotted line in Fig. 2b), suggesting cartwheel like molecules are more likely to scatter than those which approach the surface as helicopters.

To check this quantitatively we combined these two observations into a simple alignment scattering model in which scattering probabilities depend only on the alignment of the impinging molecules. We denote the equal scattering probabilities of both helicopter states ($m_J = 1$, −1) as $P_H$ and that of the cartwheel ($m_J = 0$) state as $P_C = \alpha P_H$ where α is the ratio of cartwheel to helicopter scattering probabilities.

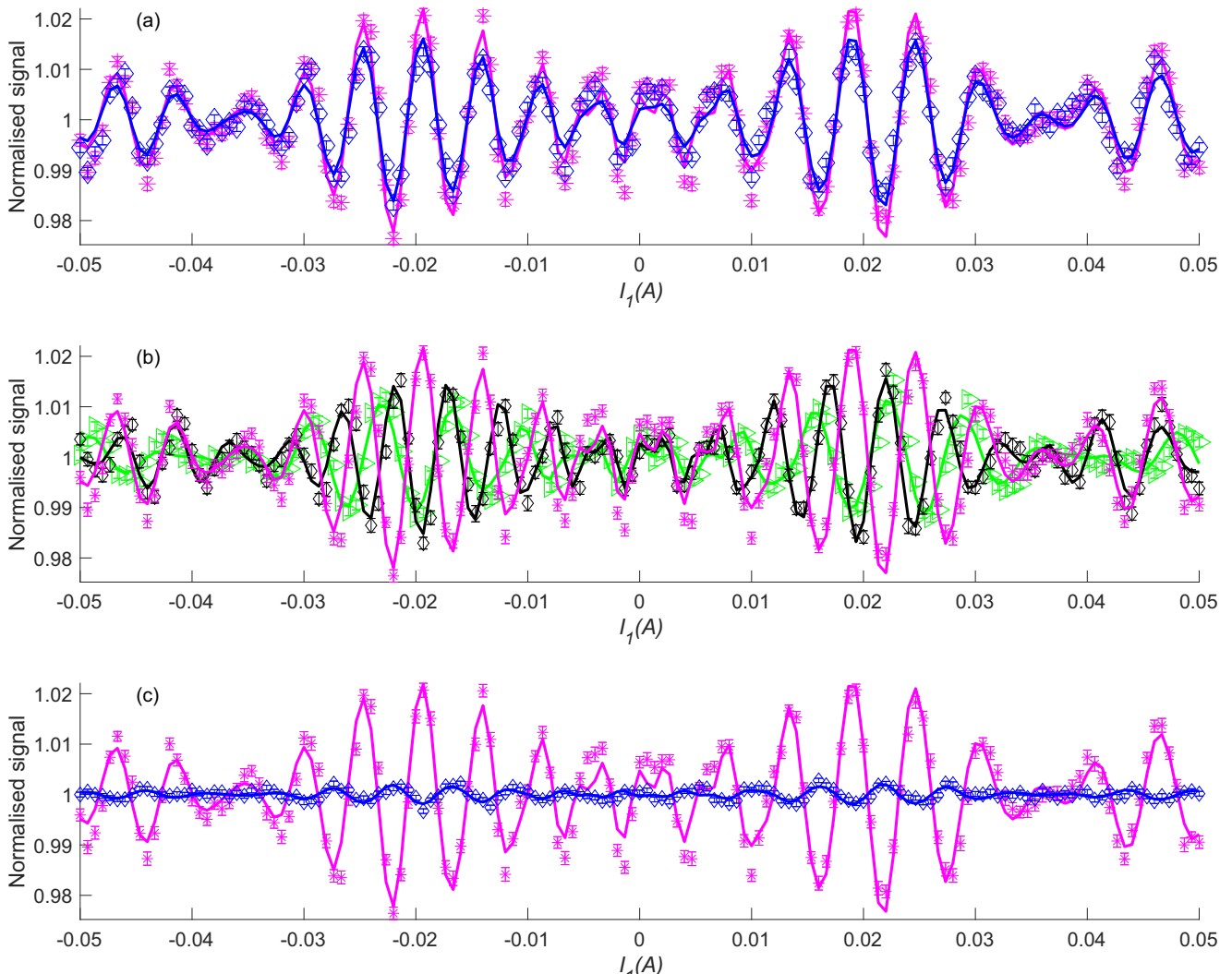

**Fig. 3 | The dependence of the specular scattering flux on the geometry, beam velocity and surface temperature. a** Comparing FD scattering measurements from a 500 K surface, using a 106 K nozzle. Magenta and blue markers and are for two different scattering geometries $\theta_{total} = 45°$ and $22.5°$. The alignment scattering model fits are shown as lines with corresponding colours. The error bars show the standard deviation of the measured values calculated from repeat measurements ($n = 61$ and $n = 200$ respectively) (**b**) Comparing FD scattering measurements from a 500 K surface for $\theta_{total} = 45°$, using a 106 K nozzle (magenta), an 84 K nozzle (black) and a 149 K nozzle (green). The fits to the alignment model are shown as lines with corresponding colours. The error bars show the standard deviation of the measured values calculated from repeat measurements ($n = 61$, 50 and 250, respectively). **c** Comparing FD scattering measurements using a 106 K nozzle and $\theta_{total} = 45°$ for a 500 K surface (magenta) which is clean and reactive to FD measured from a 180 K surface covered with a passivating layer (blue). The alignment scattering model fits are shown as lines with corresponding colours. The error bars show the standard deviation of the measured values calculated from repeat measurements ($n = 61$ and $n = 250$, respectively). Source data are provided as a Source Data file.

Within this model the FD signal can be written as

$$FD \propto \frac{P_H N_1(I_1) + P_H N_{-1}(I_1) + P_C N_0(I_1)}{< P_H N_1(I_1) + P_H N_{-1}(I_1) + P_C N_0(I_1) >}$$
$$= \frac{N_1(I_1) + N_{-1}(I_1) + \alpha N_0(I_1)}{< N_1(I_1) + N_{-1}(I_1) + \alpha N_0(I_1) >} \qquad (1)$$

where $N_1$, $N_{-1}$ and $N_0$ are the populations in the $m_J = 1$, $-1$ and 0 states as a function of the current, as shown in Fig. 2b. We normalised the model to its average value along the entire $I_1$ range, similarly to what is done for the experimental data. The thick magenta line in Fig. 2a shows the model for the best fit enhancement factor $\alpha = 1.5$, which agrees remarkably well with the experimental data, especially considering there is only one free parameter in this comparison. It should be noted that due to polarisation losses in our beam and an additional small contribution from para-$H_2$ molecules in our beam (estimated as 5%)

there is a background contribution to the measured signal. As a result the value of $\alpha$ we quote should be treated as a lower limit, i.e., the scattering probability is at least a factor of 1.5 higher for cartwheel molecules in comparison to helicopter molecules.

The fact that the alignment scattering model follows the specularly scattered signal almost perfectly, and shows an enhanced scattering for cartwheel molecules, is a rather robust property of this system, as shown in Fig. 3a, b. Figure 3a compares FD measurements for a total scattering angle of 45° (magenta markers, identical to those shown in Fig. 2a) with those measured for a total scattering angle of 22.5° (blue markers). The pattern is similar, and again the comparison with the model (blue line) is very good with a slightly higher enhancement factor ($\alpha = 1.7$). Figure 3b compares the results shown earlier in Fig. 2a (magenta asterisks) with results obtained with nozzle temperatures of 84 K (black diamonds) and 149 K (green triangles) which correspond to mean beam velocities of 1344 ms$^{-1}$ and 1786 ms$^{-1}$,

respectively (see supplementary note 1 and supplementatry Fig. 2), and which represent the energy range over which the permanent magnetic elements of the hexapole produce a significant enough population bias to be able to perform these experiments. The change of the signal patterns follows the change in the population control pattern as evidenced by the fact that the alignment scattering model (green and black lines in Fig. 3b) produce excellent fits to the data. For 84 K we get a very similar enhancement factor ($\alpha = 1.4$) and for 149 K we obtain a lower enhancement factor of $\alpha = 1.2$. We have also performed FD measurements for 3 different crystal azimuths in between [10$\bar{1}$] and [11$\bar{2}$], which all produce results which are identical within the experimental noise and are shown in Supplementary Fig. 1.

A completely different situation arises when we cool the surface (blue diamond markers in Fig. 3c), with the FD oscillations reducing by about an order of magnitude and reversing in polarity, i.e., helicopter molecules have a very small preference for scattering (best fit $\alpha = 0.97$, shown by the blue line in Fig. 3c). To understand the low surface temperature results we need to consider the dynamics of $H_2$ dissociation on this surface. When we expose the 500 K nickel surface to our molecular beam, hydrogen molecules are constantly dissociating and sticking to the surface, which considering the low velocity of our beam suggests a barrier to dissociation which is either low or non-existent.

However, at 500 K the desorption rate is higher than the adsorption rate which means that the adsorbed H atoms quickly recombine and desorb from the surface, leaving an essentially clean surface for the molecules we measure, which are those that didn't dissociate and scattered elastically towards the detector. When Ni(111) is cooled below approx. 400 K, the desorption and adsorption rates balance and H atoms remain on the surface with a coverage which depends on the flux of the molecular beam and on the surface temperature. When exposing a surface held at 180 K to our beam, a complete saturation coverage is created and the surface becomes essentially inert. The temperature dependence of the hydrogen adsorption we observe agrees with previous adsorption studies on this system[34,35].

The preferential cartwheel scattering observation, which we see when the surface is clean and reactive but disappears for the inert cold surface, could potentially be linked with stereodynamic trends of the dissociation reaction probability. Since scattering can, to a certain extent, be seen as the complementary channel to dissociation[36], enhanced scattering of cartwheel molecules could be linked with an enhanced reaction probability of helicopter molecules, as has been predicted from calculations of hydrogen reacting with other surfaces[18,19]. However, our observations can also be linked to opposing trends in other scattering channels which will be at the expense of the specular scattering probability.

In order to test whether there is a stereodynamic trend for reactive collisions, we need to combine our ability to control the rotational projection populations of the impinging molecules with a measurement that is sensitive to the coverage of H atoms on the surface. Low energy helium scattering is one of the most sensitive techniques for measuring particularly dilute adsorbate coverages[37]. To perform the measurement, we seeded the $H_2$ beam with 10% helium ($^4$He). The majority gas ($H_2$) reacts with the surface and can be stereodynamically controlled by changing the current in the solenoid, whereas the helium atoms in the beam are used to monitor the surface coverage through its effect on the reflectivity of the surface. Figure 1b illustrates this mode of measurement where two values of $I_1$ were chosen to maximise the difference between the calculated helicopter/cartwheel populations. Since the helium atoms ($^4$He) have a zero nuclear spin and are completely unaffected by magnetic fields, any change in their scattered intensity can only reflect a change of the surface itself, i.e., the number of H atoms present on the surface.

To be able to follow changes in the surface coverage we set the surface temperature to 375 K. As discussed in more detail in supplementary note 2, at this temperature equilibrium is achieved between adsorption from the $H_2$ beam and thermal desorption for a surface coverage of approximately 0.12 monolayers (ML), a coverage which reduces the helium reflectivity by 37.5 ± 1.5% in comparison to the clean Ni(111) surface. Figure 4a shows the scattered helium signal within a 170 s window, where the control solenoid current was set to abruptly change between $I_1 = -0.018$ A and $I_1 = -0.0207$ A as shown in Fig. 4b, values which were chosen to minimise and maximise the helicopter populations. We note that adding 10% helium to the beam slowed the $H_2$ molecules by approximately 100 ms⁻¹, which shifts the two population control currents by 0.0013 A (supplementary note 1 and Supplementary Fig. 3) with respect to the values for a pure $H_2$ beam. The measurements shown in Fig. 4a were repeated 150 times to obtain sufficient signal to noise and to calculate the standard deviation of the measured values. The helium signal, which was normalised to its average value, follows the magnetic manipulation sequence, decreasing when we enrich the beam with helicopters (68%) and increasing when we reduce the helicopter population (59%). Figure 4c shows an average of the measurement points at each of the two currents, excluding the first 8 s after changing the current $I_1$ to allow the surface coverage to equilibrate. The helium reflectivity changes by -1% between the two $I_1$ values.

The helium reflectivity results shown in Fig. 4, show that our magnetic manipulation controls not only the $H_2$ scattering channel but also the reaction channel, i.e., enriching the $H_2$ beam with more helicopter states (68% instead of 59%) leads to a larger yield of reaction products, i.e., more H atoms adsorbed on the surface. To quantify the difference in sticking probability we relate the changes in the helium reflectivity to the surface coverage of H atoms (see supplementary note 3 for details) and obtain an estimation of 1.2 for the minimum ratio between the sticking coefficients of helicopter and cartwheel molecules.

In the past, when interpreting experimental observations (e.g., refs. 6,7,20,38–40), a simple mutually exclusive relationship was often assumed between dominant steering of the molecules and steric preferences of adsorption, i.e., if the reaction probabilities were measured to be independent of the alignment of the incoming molecules, it was interpreted as a sign that steering effects dominate the reaction mechanism and erase any steric dependencies, and in contrast, if steric dependencies were experimentally observed, these were interpreted as a sign that steering effects are not dominant.

One possible interpretation for the anti-correlated steric trends we measured for scattering and reaction rates, challenges the mutually exclusive relationship mentioned above. We start by assuming that the potential energy surface contains both early (further from the surface) and late (closer to the surface) energy barriers, which is consistent with some of the potential energy surfaces calculated for this system[15]. If a significant number of molecules are scattered back into the gas phase at the early barrier, and if the scattering probability at the early barrier depends on the rotational orientation of the beam, the amount of molecules which will continue towards the surface, encounter the late barrier, and have some chance of eventually dissociating, will depend significantly on the rotational orientation populations of the molecular beam used in the experiment. At this point, even if strong steering occurs as molecules approach the late barrier, and even in the extreme case where all the molecules are eventually oriented towards a particular alignment where dissociation is more likely, we still expect to measure changes in the reaction probability as we alter the rotational orientation population composition of the molecular beam, due to the steric selectivity of the scattering event at the early barrier.

Calculating $m_J$ resolved scattering probabilities is particularly challenging, and even state of the art theoretical methods which have been successful in explaining trends of sticking measurements, are still not capable of calculating accurate enough potential energy surfaces to reproduce $m_J$ resolved scattering measurements (e.g., refs. 26,28). Hence, calculations which show strong steering of the molecules towards the preferred reaction geometry, and especially those which would not be considered as accurate enough to calculate the steric

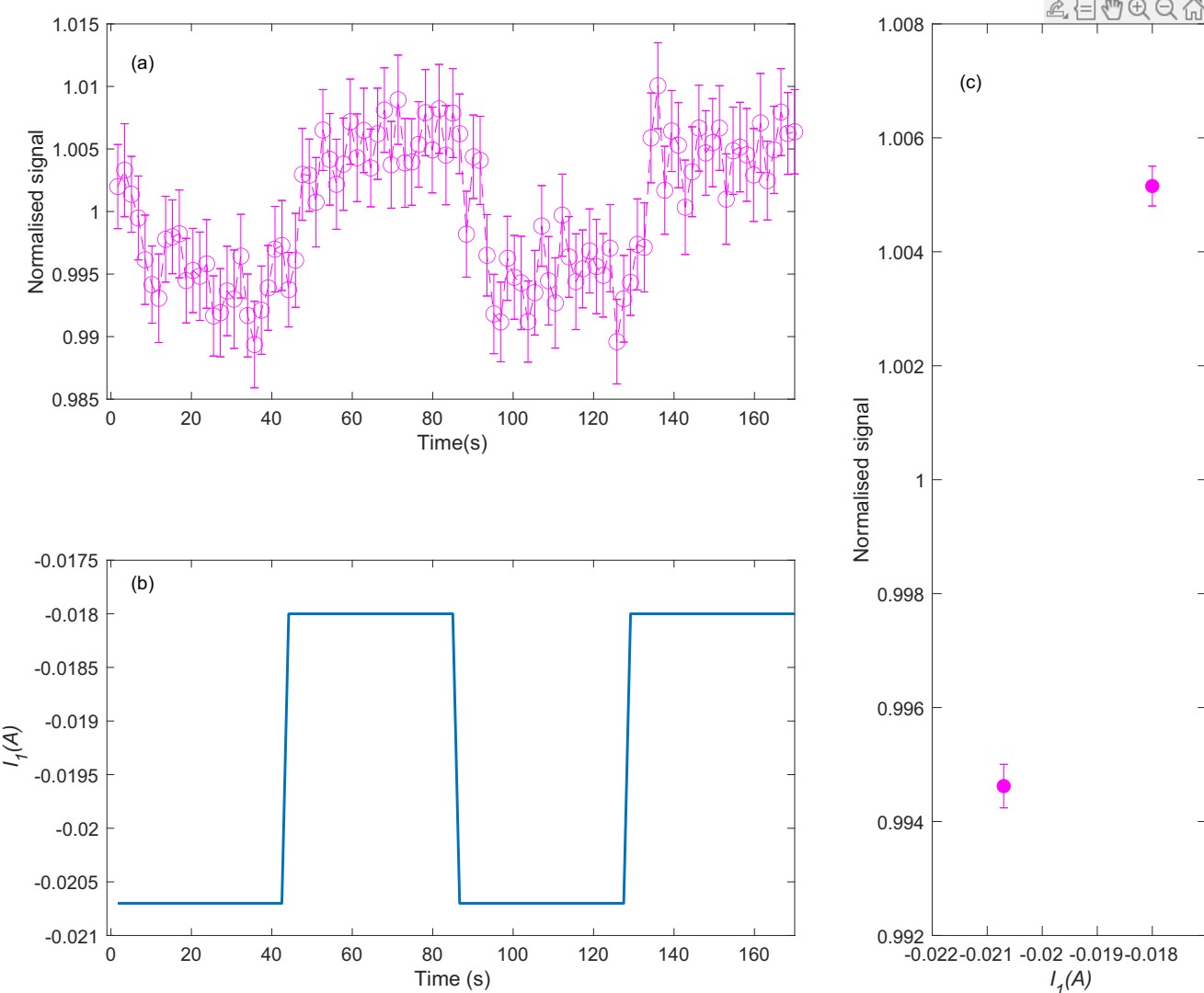

**Fig. 4 | Stereodynamic control of H₂ sticking, monitored by helium scattering.**
**a** Normalised helium signal as a function of time. The error bars represent ±1 standard deviation of the average, calculated from the repeat measurements ($n = 153$). **b** Solenoid control current, $I_1$, as a function of time. **c** Average helium signals for the two control currents, excluding the first 8 s after changing the current to let the surface coverage stabilise. The error bars represent ±1 standard deviation of the average, calculated from the repeat measurements ($n = 40$). Source data are provided as a Source Data file.

dependency of early scattering events, should not be interpreted as a counter indication for observing significant steric dependencies in dissociative sticking experiments. It is important however to stress that the existence of stereodynamic preferences in one scattering channel does not necessarily mean it will affect the observed reaction in the simple way we described above. If other scattering channels (e.g., diffraction channels and inelastic scattering events) are important and have opposite stereodynamic trends, they can also affect the number of molecules available for reaction. Further stereodynamic resolved scattering measurements, performed at different scattering geometries would be needed to account for the effects of other elastic and inelastic scattering channels.

Finally, it is interesting to consider the observations of this study within the context of previous experiments which probed the stereo-dynamics of hydrogen dissociation on surfaces. More specifically, alignment of desorbing D₂ molecules was observed from both Pd(100) and Cu(111) surfaces, from which it was deduced helicopters have a higher probability to dissociate in the reverse reaction[20,21], which is a similar trend to that shown in Fig. 4. Hou et al.[20], who measured the velocity dependence of the alignment of the desorbing molecules

from Cu(111), discussed their results in the context of two competing mechanisms, steering which is expected to reduce the alignment effects for slower molecules, and the excess of kinetic energy above the dissociation barrier, which will reduce alignment effects for faster moving molecules. The observation that the alignment increased significantly for slower molecules was interpreted as an indication that steering does not play a large role at the collision energies probed in their study[20]. A quantitative comparison with the results of this study would be meaningless due to the completely different measurement techniques, energy barriers to dissociation, rotational states and kinetic energy of the beam particles. However, in the light of the anti-correlated stereodynamics trends for scattering and dissociation we measured on Ni(111), and the general possibility that early scattering events can lead to observed stereodynamic trends in reaction rates, perhaps one conclusion of this study is to keep an open mind about the existence of significant steering of H₂ on any of these surfaces. Once theoretical modelling is accurate enough to reproduce $m_J$ state-resolved measurements of both scattering and reaction rates, it could be used to provide valuable insight into the fine details of the potential energy surface and the dissociation mechanism.

## Methods

Details of the MMI apparatus can be found in previous publications[26,27], below we provide details for the elements which are specific to this study. The Ni(111) sample (Surface Preparation Lab, The Netherlands) was mounted on a home-built, 6-axis manipulator in the ultra-high vacuum chamber at the end of the first arm of the apparatus. The surface was cleaned using repeated sputter-anneal cycles, where sputtering was performed by exposing a surface held at a temperature of 500 K to a ~ 12 μA beam of 1 KeV argon ions for 10 min, annealing was performed by heating the sample to 800 K for 15–30 s, and the cycles were repeated until the specular scattering signal stopped improving. The temperature of the sample was monitored using a T-type thermocouple with an absolute error estimated as ±0.75 K. The surface azimuth orientation was verified to within ±1° using the helium diffraction pattern. Since nickel is a ferromagnet, it could potentially be magnetised and affect the results and the interpretation of the data. We performed control experiments with a beam of $^3$He atoms (740 ms$^{-1}$) to assess this possibility. The oscillation patterns measured for helium scattering from Cu(111) and Ni(111) were compared and appeared identical within the experimental noise, indicating the absence of any surface magnetism which is strong enough to affect the magnetic moments of the beam particles.

Two differentially pumped mass spectrometers at different scattering angles were used to detect the scattered $H_2$ beam. The first is an RGA-200 (Standford Research Systems) connected to a port on the UHV chamber, at a total scattering angle of 22.5° (corresponding to an incoming and outgoing angle of 11.25° for the specularly scattered measurements presented here). The second is a HAL-201 (Hiden) which can be moved into the beamline after the solenoid and before the hexapole in the second arm of the MMI apparatus, which detects signal scattered through a total scattering angle of 45° (22.5° incoming and outgoing angle for specular scattering). We note that both the RGA-200 and the HAL-201 were operated in stagnation mode, i.e., they measure the increase in the equilibrium partial pressure, which is proportional to the flux of the molecular beam entering the chambers they are located in. To monitor the scattered helium signal for the reactivity measurement, a Max120 (Extrel) mass spectrometer was used which was also positioned at a total scattering angle of 45° but offered higher sensitivity.

To obtain the $m_J$ state populations of the molecules that collide with the sample, it is necessary to calculate the propagation of the 9 $m_I, m_J$ states of $H_2$ through the magnetic components of the beamline shown schematically in Fig. 1a of the main manuscript. The first step is to calculate the probabilities, $P_{hex}(m_I, m_J)$, that each $m_I, m_J$ is transmitted through the hexapole polariser. To do this we use a semi-classical trajectory calculation where the motion of the molecule is propagated classically, but the (quantised) forces are calculated according to their $m_I, m_J$ state[41]. Due to the strong magnetic field gradients in the hexapole, the superposition states decohere[42] leaving the molecules in 1 of the 9 pure $m_I, m_J$ states at the end of the hexapole with an unequal population distribution.

The propagation of the quantum states from the end of the hexapole to the surface is performed semi-classically[26], with the motion of the centre of mass of the molecule calculated classically but the evolution of the $m_I, m_J$ states quantum mechanically, using the Hamiltonian for the $J = 1, I = 1$ state of $H_2$[30]. The solution of the evolution can be written as a propagation matrix, $U(I_1)$, which expresses the superposition state obtained from an initially pure $m_I, m_J$ state after it was propagated through the magnetic field profile associated with a given solenoid current, $I_1$. The relative population in a given $m'_J$ state when the molecules reach the surface, $\Omega(m'_J)$, can then be calculated by projecting the wavefunction of the molecule on to the $m'_I, m'_J$ state at the surface position (where the quantisation axis is taken to be the surface normal) and taking the square modulus before summing over the final $m_I$ states, velocity distribution ($P_v$) and hexapole transmission probabilities, $P_{hex}(m_I, m_J)$, i.e.,

$$\Omega(m'_J) = \sum_v P_v \sum_{m_I, m_J, m'_I} P_{hex}(m_I, m_J) |\langle m'_I, m'_J | R(\theta) U(I_1) | m_I, m_J \rangle|^2 \quad (2)$$

where $R(\theta)$ is the rotation matrix that changes the quantisation axis from the direction of the dipole at the end of the hexapole to the surface normal.

## Data availability

The authors declare that the data supporting the findings of this study are available within the paper and its Supplementary Information files. Should any raw data files be needed in another format they are available from the corresponding author upon request. Source data are provided with this paper.

## Code availability

The Matlab scripts used to plot the measurements and the alignment model fit are available at. https://github.com/GilAlexandrowicz/Codes---Rotational-orientation-control-.git.

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

## Acknowledgements

The authors thank Geert-Jan Kroes and Mark Somers for valuable discussions and acknowledge the support of the Supercomputing Wales project, which is part-funded by the European Regional Development Fund (ERDF) via Welsh Government. This project was funded by a UKRI, Future Leader Fellowship MR/X03609X/1 (H.C.) and an EPSRC, grant EP/X037886/1 (G.A., H.C.).

## Author contributions

H.C. performed the dissociation measurements, G.Z. and C.J.B. performed the scattering measurements. H.C. calculated the rotational projection populations. P.L.S. adapted the MMI apparatus to perform the flux detection measurements. G.A. and H.C. interpreted the measurements and wrote the manuscript. G.A. conceived and supervised the project and developed the control methodology. All the authors read and commented on the manuscript.

## Competing interests

The authors declare no competing interests.
