## [Transparent Peer Review file · Nature Communications]

Rotational orientation control of ground-state ortho-H₂ dissociation on a metal surface.

Corresponding Author: Professor Gil Alexandrowicz

Version 0:

Reviewer comments:

Reviewer #2

(Remarks to the Author)
See attachment.

Reviewer #3

(Remarks to the Author)

As stated in my previous review (reviewer #3) of this paper for Nature Chemistry, I believe the manuscript reports on great results on controlling dissociative adsorption of H₂ on nickel by rotational alignment. Due to the lack of theoretical calculations, I suggested either adding theoretical support or submitting a revised manuscript to a more specialized journal. The authors have decided to submit to a more specialized journal by transferring the manuscript to Nature Communications, which I believe is a good fit for the manuscript. The authors have satisfactorily addressed most of my comments and I suggest to accept the manuscript with minor revisions. Below are remarks on two of the responses of the authors, an additional comment, and some minor comments that the authors might want to consider.

Concerning my comments about theoretical calculations:

Ultimately, the main motivation of such experiments is to gain a deep (or deeper) understanding of the underlying dynamics. In my opinion, the lack of theoretical modeling means that this goal is not fully achieved, making it less interesting for a broad audience. Still the experiment marks an important step in this direction and will for sure motivate theoretical work in the future. Nevertheless, I believe that the data presented would allow some additional statements to be made about the system. For example, one could say that the system has a very low or no barrier to adsorption. Furthermore, the results could be compared to the literature and it could be discussed, if the new results are in agreement with previous work (see also previous comments of reviewer #2). In my opinion, such an extended discussion would strengthen the paper and emphasize the value of the reported data. However, I would like this to be understood as a suggestion.

Concerning my comment about the novelty of the experimental method:

Of course, the authors are absolutely right that the use of a He seeded beam to monitor the surface coverage and to directly measure the anti-correlation between scattering and sticking is a significant experimental development. However, the way the manuscript is written, it sounds to me more like the method to control the rotational orientation of the molecules is new. Adding a few sentences or even a paragraph in the introduction (page 3, line 69-78) about this new approach would help to clarify this. Currently, this new approach is first mentioned very late in the paper, on page 7, line 240-252. I think this clever idea should already be highlighted in the introduction.

I am having trouble following the discussion in paragraph 1 on page 9 (line 295-314). Is the essence of this paragraph that cartwheel molecules see an "earlier barrier" and do not get close enough to experience steering forces, while helicopter molecules see a "later barrier" and get close enough to the surface to dissociate? I am not sure what the intent of this discussion is. Are the authors assuming that steering effects should be important at the experimental conditions used, and

are trying to argue why alignment effects are still observed? I would ask the authors to sharpen this paragraph and make the intended argument clear.

Minor Comments:

While the speed of the 84 K and 149 K beams is given in the main text, the speed of the 106 K beam is given only in the SI. It should also be given in the main text, either in the caption of Figure 2 or in the paragraph where the other speeds are given (page 6, line 205).

Page 7, line 233: Why is diffractive scattering used here? Shouldn't that be specular scattering or just scattering?

Page 7, line 260: Maybe one could mark both currents in figure S3 in the SI?

I do not believe that the statement in the last sentence of the manuscript is true (page 9, line 327-329: However, the fact that the scattering stereodynamic trends do not change between 375K and 600K implies that the static surface approximation can be safely used). Every reactive encounter will require the surface atoms to move to their new equilibrium position. Even the transition state for dissociation may be a different structure than the equilibrium structure. As stated in the cited reference 40, the static surface approximation can only achieve qualitative agreement and predict trends. However, I believe that a static surface approximation will not be sufficient to model the reported experiments.

Version 1:

Reviewer comments:

Reviewer #2

(Remarks to the Author)

In the revised version of the manuscript, the authors have addressed all of my previous comments and concerns satisfactorily. In particular, the revisions to the discussion section around Page 10 are a significant improvement. I understand much better now the argument that the authors put forward, and what they believe might be the possible mechanisms underlying their observations. The authors have also added some discussion about the state of the art in theoretical calculations of mJ-resolved scattering distributions, which provide context for the reader. The new last paragraph of the paper also improves the discussion, providing information about what comparisons can and cannot be made to prior desorption measurements. I think this discussion actually strongly motivates the need for experiments that probe the forward dissociative chemisorption reaction, because it provides different, complementary information to recombinative desorption experiments. The authors also clarified the experimental question I had about flux detection—it is clear in the revised version. I do not have any further comments. I think this is a well-written paper that describes important, one-of-a-kind experiments that probe a crucially important chemical problem, providing significant new insight.

Reviewer #3

(Remarks to the Author)

The authors have addressed all my comments to my satisfaction, and I suggest to accept the paper for publication in Nature Communications in its current form.

Response to comments / suggestions of the reviewers.

Original reviewer comments are in black fonts, our responses and the changes we made are written in blue and red fonts correspondingly.

Reviewer 2

We thank the reviewer for their comments and efforts reviewing this paper.

1. I do not understand what the authors are getting at with the discussion in the paragraph starting on line 295. They argue that strong steering effects could be present, but not at the distances at which “cartwheel” molecules undergo specular scattering. Isn’t that the same thing as arguing that steering effects are not strong enough or long-range enough to wash out the stereodynamic preference for helicopter molecules to have a larger sticking probability than cartwheel molecules? I also do not understand the logic of the first sentence of that paragraph, which states, “The anti-correlated steric trends we measured for scattering and reaction point to the possibility of a more complex relationship for dissociative adsorption, where both alignment effects and steering can coexist.” Maybe I am missing something that the authors could clarify to me, but I don’t see why the observed anti-correlation implies such a complicated relationship. (I do not disagree that there could be a complicated interplay, but I do not see how it is implied by the anti-correlation.) Doesn’t the anti-correlation simply imply the straightforward conclusion that, under the conditions investigated, cartwheel molecules are more likely to undergo specular scattering, whereas helicopter molecules are more likely to stick and therefore not scatter?

The anti-correlation of the type we measured could imply a causation relation between the two events, although possibly not of the type the reviewer might have in mind “helicopter molecules are more likely to stick and therefore not scatter”, since the low sticking probability (<0.05) means that that the sticking cannot be the process which is driving the differences between the scattering probabilities of helicopter and cartwheels molecules. The explanation we think is more likely is the existence of an early scattering event which is more effective in scattering cartwheel molecules back into the gas phase. This means reactions, which take place at a late barrier (i.e. closer to the surface) will happen more frequently for helicopter rich beams (as less molecules in cartwheels rich beams will arrive at the late barrier), regardless of any steering which takes place closer to the surface. Following the feedback from both reviewers, we have modified the text in a way which hopefully clarifies this point.

Changes to the manuscript. We have modified 3 paragraphs on the last page of the paper which now read

“In the past, when interpreting experimental observations (e.g. ^{6,7,20,38–40}), a simple mutually exclusive relationship was often assumed between dominant steering of the molecules and steric preferences of adsorption, i.e. if the reaction probabilities were measured to be independent of the alignment of the incoming molecules, it was interpreted as a sign that steering effects dominate the reaction mechanism and erase any steric dependencies, and in contrast, if steric dependencies were experimentally observed, these were interpreted as a sign that steering effects are not dominant.

One possible interpretation for the anti-correlated steric trends we measured for scattering and reaction rates, challenges the mutually exclusive relationship mentioned above. We start by assuming that the potential energy surface contains both early (further from the surface) and late (closer to the surface) energy barriers, which is consistent with some of the potential energy surfaces calculated for this system¹⁵. If a significant number of molecules are scattered back into the gas phase at the early barrier, and if the scattering probability at the early barrier depends on the rotational orientation of the beam, the amount of molecules which will continue towards the surface, encounter the late barrier, and have some chance of eventually dissociating, will depend significantly on the rotational orientation populations of the molecular beam used in the experiment. At this point, even if strong steering occurs as molecules approach the late barrier, and even in the extreme case where all the molecules are eventually oriented towards a particular alignment where dissociation is more likely, we still expect to measure changes in the reaction probability as we alter the rotational orientation population composition of the molecular beam, due to the steric selectivity of the scattering event at the early barrier.

Calculating m_j resolved scattering probabilities is particularly challenging, and even state of the art theoretical methods which have been successful in explaining trends of sticking measurements, are still not capable of calculating accurate enough potential energy surfaces to reproduce m_j resolved scattering measurements (e.g. ^{26,28}). Hence, calculations which show strong steering of the molecules towards the preferred reaction geometry, and especially those which would not be considered as accurate enough to calculate the steric dependency of early scattering events, should not be interpreted as a counter indication for observing significant steric dependencies in dissociative sticking experiments. It is important however to stress that the existence of stereodynamic preferences in one scattering channel does not necessarily mean it will affect the observed reaction in the simple way we described above. If other scattering channels (e.g. diffraction channels and inelastic scattering events) are important and have opposite stereodynamic trends, they can also affect the number of molecules available for reaction. Further stereodynamic resolved scattering measurements, performed at different scattering geometries would be needed to account for the effects of other elastic and inelastic scattering channels.”

2. I think the manuscript could be stronger if there were some discussion about the underlying mechanism. In Ref. 20, an argument was presented that the observation was inconsistent with steering because the alignment of desorbed molecules became stronger as the velocity decreased. It was argued that this observation suggests that alignment becomes less important as the incidence energy in excess of the barrier increases. In the current work, a similar trend is observed (the enhancement factor α decreases somewhat when TN is increased from 106 K to 149 K). Do the authors think this trend is significant or is it not strong enough to merit comment? Based on the discussion, it sounds like the authors are surprised that steering appears to be absent, even at these low incidence energies. However, other systems like H₂/Cu have larger barriers, and it has been argued that what counts is excess energy above the barrier (i.e. how much kinetic energy remains after deceleration). In my opinion, an insightful discussion about the incidence energy dependence of steric effects in different systems with different barrier heights and shapes could improve the manuscript.

First, we would like to stress that we do not intend to claim the data proves steering is absent and correspondingly we are not surprised by its absence. In fact we were trying to make the point that the fact we see steric dependencies in the reaction does not rule out steering.

Hopefully the explanations given to the previous point and the changes to the manuscript in the last section of the paper will clarify this point.

Changes to the manuscript. We have modified 3 paragraphs on the last page as detailed in the response to the previous comment. We have also removed the word “particularly” in the abstract when referring to the possibility of the steering being weak, to avoid misleading the reader into thinking that this would be surprising.

Regarding the velocity dependence: In ref 20 the property which is discussed is the velocity dependence of the stereoselectivity of dissociative adsorption inferred from a state selective desorption measurement. The change in α as a function of molecular beam velocity mentioned by the reviewer describes changes in the stereoselectivity of the specular scattering channel. While the two processes (specular scattering and reaction) might be closely related (see discussion above), we believe an additional discussion relating the velocity dependence of the scattering stereo selectivity we see between 106K to 149K to the unknown barrier height (which is indeed a very interesting question) is quite speculative at this stage, and we would prefer to not include it in the manuscript. Having said this, we did add a final paragraph which considers the observations of this study within the context of previous experiments and in particular the paper mentioned by the reviewer.

New text (last paragraph of the paper) :

“Finally, it is interesting to consider the observations of this study within the context of previous experiments which probed the stereodynamics of hydrogen dissociation on surfaces. More specifically, alignment of desorbing D₂ molecules was observed from both Pd(100) and Cu(111) surfaces, from which it was deduced helicopters have a higher probability to dissociate in the reverse reaction^{20,21}, which is a similar trend to that shown in figure 4. Hou et al.²⁰, who measured the velocity dependence of the alignment of the desorbing molecules from Cu(111), discussed their results in the context of two competing mechanisms, steering which is expected to reduce the alignment effects for slower molecules, and the excess of kinetic energy above the dissociation barrier, which will reduce alignment effects for faster moving molecules. The observation that the alignment increased significantly for slower molecules was interpreted as an indication that steering does not play a large role at the collision energies probed in their study²⁰. A quantitative comparison with the results of this study would be meaningless due to the completely different measurement techniques, energy barriers to dissociation, rotational states and kinetic energy of the beam particles. However, in the light of the anti-correlated stereodynamics trends for scattering and dissociation we measured on Ni(111), and the general possibility that early scattering events can lead to observed stereodynamic trends in reaction rates, perhaps one conclusion of this study is to keep an open mind about the existence of significant steering of H₂ on any of these surfaces. Once theoretical modelling is accurate enough to reproduce m_J state-resolved measurements of both scattering and reaction rates, it could be used to provide valuable insight into the fine details of the potential energy surface and the dissociation mechanism.

”

Minor Comments

1. The authors refer to their current detector configuration as “flux detection (FD)”. However, to my knowledge the residual gas analyzers used in this work usually give a

signal that is proportional to the density of molecules in the electron impact ionization region—not the flux of molecules passing through the region. If the speed distribution is narrow, the two quantities will be approximately proportional, but I find the terminology confusing. If I have misunderstood the regime in which the detection operates, perhaps the authors could provide an explanation.

This is a good point, but we think the reviewer might have missed the answer to this comment in our previous rebuttal letter, explaining that the pressure was measured in stagnation mode rather than in a fly through configuration and the changes we made to the text. Just to be sure we have now added a second clarification to make this point clearer.

Changes introduced in the last version of the manuscript:

The word stagnation was added (2nd paragraph of page 3), the text reads : “For this work we modified the MMI setup, adding two differentially-pumped mass spectrometers at two different angles which allow us to measure the stagnated pressure of the scattered beam directly...”

Additional change in this version.

We have added a more detailed explanation in the experimental methods section which explains what we mean by stagnated pressure. The text which was added:

“We note that both the RGA-200 and the HAL-201 were operated in stagnation mode, i.e. they measure the increase in the equilibrium partial pressure, which is proportional to the flux of the molecular beam entering the chambers they are located in.”

2. I noticed one typo: on page 10, line 343 it should say “affect”, not “effect”.

We thank the Reviewer for highlighting this.

Changes: The typo has been corrected.

3. Since the journal typically does not do any copy editing to the SI, it would be helpful if the authors could clean up some of the typesetting. For example, on the third line of page 2 of the SI, the prime symbols on mI' and mJ' are difficult to read because they typesetting causes them to appear too far above the symbol. On the fourth line, the apostrophe symbol should be replaced by a prime symbol.

We have been through the SI as requested and changed the typesetting and apostrophe symbol as highlighted by the Reviewer.

Reviewer #3 (Remarks to the Author):

As stated in my previous review (reviewer #3) of this paper for Nature Chemistry, I believe the manuscript reports on great results on controlling dissociative adsorption of H₂ on nickel by rotational alignment. Due to the lack of theoretical calculations, I suggested either adding theoretical support or submitting a revised manuscript to a more specialized journal. The

authors have decided to submit to a more specialized journal by transferring the manuscript to Nature Communications, which I believe is a good fit for the manuscript. The authors have satisfactorily addressed most of my comments and I suggest to accept the manuscript with minor revisions. Below are remarks on two of the responses of the authors, an additional comment, and some minor comments that the authors might want to consider.

We thank the reviewer for their comments and efforts reviewing this paper.

Concerning my comments about theoretical calculations:

Ultimately, the main motivation of such experiments is to gain a deep (or deeper) understanding of the underlying dynamics. In my opinion, the lack of theoretical modeling means that this goal is not fully achieved, making it less interesting for a broad audience. Still the experiment marks an important step in this direction and will for sure motivate theoretical work in the future. Nevertheless, I believe that the data presented would allow some additional statements to be made about the system. For example, one could say that the system has a very low or no barrier to adsorption. Furthermore, the results could be compared to the literature and it could be discussed, if the new results are in agreement with previous work (see also previous comments of reviewer #2). In my opinion, such an extended discussion would strength the paper and emphasize the value of the reported data. However, I would like this to be understood as a suggestion.

We agree with the suggestion to comment on the barriers being low or non-existent. In terms of comparing with existing literature about this system; quantitative comparisons can be made in terms of the temperature dependent adsorption properties. This comparison already existed in the supplementary information section but we have now also added a remark about this in the main text. Regarding the suggestion of reviewer 2 to include a comparison with velocity dependent effects seen in reference 20, please see our response earlier in this document, and the changes made to the manuscript (last paragraph of the modified manuscript).

Changes: first full paragraph in page 7 now reads:

A completely different situation arises when we cool the surface (blue diamond markers in figure 3c), with the FD oscillations reducing by about an order of magnitude and reversing in polarity, i.e. helicopter molecules have a very small preference for scattering (best fit $\alpha = 0.97$, shown by the blue line in figure 3c). To understand the low surface temperature results we need to consider the dynamics of H₂ dissociation on this surface. When we expose the 500K nickel surface to our molecular beam, hydrogen molecules are constantly dissociating and sticking to the surface, which considering the low velocity of our beam suggests a barrier to dissociation which is either low or non-existent.

At 500K the desorption rate is higher than the adsorption rate which means that the adsorbed H atoms quickly recombine and desorb from the surface, leaving an essentially clean surface for the molecules we measure, which are those that didn't dissociate and scattered elastically towards the detector. When Ni(111) is cooled below approx. 400K, the desorption and adsorption rates balance and H atoms remain on the surface with a coverage which depends on the flux of the molecular beam and on the surface temperature. When exposing a surface held at 180K to our beam, a complete saturation coverage is created and the surface becomes essentially inert. The temperature dependence of the hydrogen adsorption we observe agrees with previous adsorption studies on this system^{34,35}, and is discussed further in the supplementary information section."

Concerning my comment about the novelty of the experimental method:

Of course, the authors are absolutely right that the use of a He seeded beam to monitor the surface coverage and to directly measure the anti-correlation between scattering and sticking is a significant experimental development. However, the way the manuscript is written, it sounds to me more like the method to control the rotational orientation of the molecules is new. Adding a few sentences or even a paragraph in the introduction (page 3, line 69-78) about this new approach would help to clarify this. Currently, this new approach is first mentioned very late in the paper, on page 7, line 240-252. I think this clever idea should already be highlighted in the introduction.

We agree with this comment and have highlighted the use of a He seeded beam to monitor the H coverage on the surface earlier in the manuscript.

Changes to the manuscript. “While the wealth of information from a FI measurement is particularly beneficial to benchmark and test theoretical calculations^{27,28}, FD measurements can be much simpler to interpret, as will become obvious from the example we show below. An additional capability we have added to the setup is the use of mixed beams, where a small quantity of helium atoms are added to the molecular beam, allowing us to monitor changes in the surface coverage while simultaneously controlling the rotational states of the impinging hydrogen molecules, experiments which will be explained and presented later in this manuscript.”

I am having trouble following the discussion in paragraph 1 on page 9 (line 295-314). Is the essence of this paragraph that cartwheel molecules see an “earlier barrier” and do not get close enough to experience steering forces, while helicopter molecules see a “later barrier” and get close enough to the surface to dissociate?

Yes, this is the general idea of what we mean. The point is that if some steric selection happens early enough due to the scattering process, a steering mechanism which takes place later in the approach, will not erase the ability to measure changes in the reaction rate when the alignment in the molecular beam is changed.

Changes in the manuscript: We have rephrased the text of page 9 (for details see response to the comments of reviewer#2).

I am not sure what the intent of this discussion is. Are the authors assuming that steering effects should be important at the experimental conditions used, and are trying to argue why alignment effects are still observed? I would ask the authors to sharpen this paragraph and make the intended argument clear.

The importance of the discussion is not to argue whether steering at the late barrier exists in this system, theory is not accurate enough to reconstruct a reliable potential energy surface for this system, and the experimental data can not single out the reaction mechanism on its own. The importance of highlighting the possibility that the reactivity trends we measured could be related to steric preferences of scattering at an early barrier, is that in the literature (and there are several examples of this), steering of the molecule towards the reaction event, and the observation of steric dependencies in alignment controlled experiments, are often treated as contradictory scenarios. Assuming a mutually exclusive relationship between the two is oversimplistic, and any suggested mechanism should take both scattering and reaction

preferences into account. In the future, when theory will be capable of calculating this system accurately enough to reproduce our measurements, it should also be able to provide insight into the detailed mechanism.

As mentioned above, we have changed the 3 paragraphs at the end of the manuscript (as detailed in the response to reviewer 2) and used the terminology suggested by the reviewer (early and late barriers) which we think clarifies the argument.

Minor Comments:

While the speed of the 84 K and 149 K beams is given in the main text, the speed of the 106 K beam is given only in the SI. It should also be given in the main text, either in the caption of Figure 2 or in the paragraph where the other speeds are given (page 6, line 205).

We agree with the reviewer.

Changes to the manuscript. We have added the information in the caption of figure 2 “The nozzle temperature was stabilised at 106K producing a molecular beam with a mean velocity of 1513 ms^{-1} and a FWHM of 7%.”

Page 7, line 233: Why is diffractive scattering used here? Shouldn't that be specular scattering or just scattering?

We agree

Changes to the manuscript: “Diffractive scattering was changed to scattering”.

Page 7, line 260: Maybe one could mark both currents in figure S3 in the SI?

We agree with the reviewer's remark.

Changes: The control currents have been added as grey lines in figure S3 (a) of the SI.

I do not believe that the statement in the last sentence of the manuscript is true (page 9, line 327-329: However, the fact that the scattering stereodynamic trends do not change between 375K and 600K implies that the static surface approximation can be safely used). Every reactive encounter will require the surface atoms to move to their new equilibrium position. Even the transition state for dissociation may be a different structure than the equilibrium structure. As stated in the cited reference 40, the static surface approximation can only achieve qualitative agreement and predict trends. However, I believe that a static surface approximation will not be sufficient to model the reported experiments.

We agree with the referee that changes in the position of the surface atoms due to the proximity of the hydrogen molecule will take place and could affect the reactivity significantly. The point we were trying to make is that since the surface temperature doesn't seem to alter the steric dependencies in any measurable way, theoretical modelling of this system without including surface temperature effects could be justified. Either way, to avoid controversial or potentially misleading claims, we decided to remove that sentence. The fact that heating the surface temperature to 600K does not change the experimental observations

is mentioned in the SI and we will leave it to the reader to decide how this affects or doesn't affect the choice of an appropriate theoretical model.

Changes to the text: The sentence mentioned above was removed from the text.

This manuscript uses a novel approach based on magnetic molecular interferometry to measure the effect of rotational alignment on the sticking coefficient of H₂ on Ni(111). Although similar stereodynamic effects were discovered over two decades ago for H₂ sticking on metal surfaces (see, e.g., Ref. 20), previous investigations relied on measuring alignment of the rotational angular momentum resulting from recombinative desorption of H₂ (or D₂) and invoking detailed balance arguments to describe rotational alignment dependence of the reverse reaction (dissociative chemisorption). To my knowledge, Chadwick *et al.* demonstrate the first experiment in which the incidence rotational orientation of a beam of H₂ in its ground vibrational state is *controlled* and used to *directly* measure the rotational orientation dependence of a H—H bond-breaking reaction. The approach is powerful because it allows fine control over incidence velocity and angle, can be used to probe non-thermal incidence energy regimes, and can probe trajectories that do *not* lead to sticking. The demonstration of this technique is significant, because it can be used to obtain much more detailed information than can be extracted from desorption measurements.

The chosen system for this study is important because the sticking of H₂ on metal surfaces like nickel plays a centrally important role in heterogeneously catalyzed hydrogenation reactions and hydrogen storage. This strongly motivates experiments like the one described here, which provide stringent benchmarks for theory. As the authors point out, theory is currently incapable of reproducing experiment across the full range of incidence conditions that have been studied.

The key results of the experiment are demonstrated very clearly in two sets of complementary experiments. The measurement of H₂ scattering (integrated over final internal quantum states) as a function of current through the solenoid on the incident beam arm clearly indicates a significantly higher likelihood of specular scattering when the incident molecules are rotating in the “cartwheel” sense rather than the “helicopter” sense. This preference is observed at two different incidence polar angles, at several incidence azimuthal angles, and across a range of average incidence velocities. In a second experiment, the surface H coverage is monitored by low energy helium scattering while the “cartwheel” to “helicopter” ratio is modulated. The surface is held at a carefully chosen temperature at which the steady-state coverage is about 20% of the saturation coverage. The modulation of the He signal indicates clearly that the sticking coefficient for the $M_J = \pm 1$ (helicopter) molecules is significantly higher than for the $M_J = 0$ (cartwheel) molecules. The explanation of the experimental results is well written.

The authors have made some revisions that addressed some of the concerns I had about a previous version of the manuscript, and they have simplified the discussion section. My assessment is that the manuscript is acceptable for publication in Nature Communications after minor revisions. Below, I have written a few questions and comments regarding the discussion.

Comments and Questions

1. I do not understand what the authors are getting at with the discussion in the paragraph starting on line 295. They argue that strong steering effects could be present, but not at the distances at which “cartwheel” molecules undergo specular

scattering. Isn't that the same thing as arguing that steering effects are not strong enough or long-range enough to wash out the stereodynamic preference for helicopter molecules to have a larger sticking probability than cartwheel molecules? I also do not understand the logic of the first sentence of that paragraph, which states, "The anti-correlated steric trends we measured for scattering and reaction point to the possibility of a more complex relationship for dissociative adsorption, where both alignment effects and steering can coexist." Maybe I am missing something that the authors could clarify to me, but I don't see why the observed anti-correlation implies such a complicated relationship. (I do not disagree that there *could* be a complicated interplay, but I do not see how it is implied by the anti-correlation.) Doesn't the anti-correlation simply imply the straightforward conclusion that, under the conditions investigated, cartwheel molecules are more likely to undergo specular scattering, whereas helicopter molecules are more likely to stick and therefore *not* scatter?

2. I think the manuscript could be stronger if there were some discussion about the underlying mechanism. In Ref. 20, an argument was presented that the observation was inconsistent with steering because the alignment of desorbed molecules became stronger as the velocity decreased. It was argued that this observation suggests that alignment becomes less important as the incidence energy in excess of the barrier increases. In the current work, a similar trend is observed (the enhancement factor α decreases somewhat when T_N is increased from 106 K to 149 K). Do the authors think this trend is significant or is it not strong enough to merit comment? Based on the discussion, it sounds like the authors are surprised that steering appears to be absent, even at these low incidence energies. However, other systems like H_2/Cu have larger barriers, and it has been argued that what counts is *excess energy* above the barrier (i.e. how much kinetic energy remains after deceleration). In my opinion, an insightful discussion about the incidence energy dependence of steric effects in different systems with different barrier heights and shapes could improve the manuscript.

Minor Comments

1. The authors refer to their current detector configuration as "flux detection (FD)". However, to my knowledge the residual gas analyzers used in this work usually give a signal that is proportional to the *density* of molecules in the electron impact ionization region—not the flux of molecules passing through the region. If the speed distribution is narrow, the two quantities will be approximately proportional, but I find the terminology confusing. If I have misunderstood the regime in which the detection operates, perhaps the authors could provide an explanation.
2. I noticed one typo: on page 10, line 343 it should say "affect", not "effect".
3. Since the journal typically does not do any copy editing to the SI, it would be helpful if the authors could clean up some of the typesetting. For example, on the third line of page 2 of the SI, the prime symbols on m_i' and m_j' are difficult to read because they typesetting causes them to appear too far above the symbol. On the fourth line, the apostrophe symbol should be replaced by a prime symbol.